# The Concept of Using LSTM to Detect Moisture in Brick Walls by Means of Electrical Impedance Tomography

Grzegorz Kłosowski [1,*], Anna Hoła [2], Tomasz Rymarczyk [3,4], Łukasz Skowron [1], Tomasz Wołowiec [5] and Marcin Kowalski [3]

1 Faculty of Management, Lublin University of Technology, 20-618 Lublin, Poland; l.skowron@pollub.pl
2 Faculty of Civil Engineering, Wrocław University of Science and Technology, 50-370 Wrocław, Poland; anna.hola@pwr.edu.pl
3 Faculty of Transport and Computer Science, University of Economics and Innovation in Lublin, 20-209 Lublin, Poland; tomasz@rymarczyk.com (T.R.); marcin.kowalski@wsei.lublin.pl (M.K.)
4 Research & Development Centre Netrix S.A., 20-704 Lublin, Poland
5 Institute of Public Administration and Business, University of Economics and Innovation in Lublin, 20-209 Lublin, Poland; tomasz.wolowiec@wsei.lublin.pl
* Correspondence: g.klosowski@pollub.pl

**Abstract:** This paper refers to an original concept of tomographic measurement of brick wall humidity using an algorithm based on long short-term memory (LSTM) neural networks. The measurement vector was treated as a data sequence with a single time step in the presented study. This approach enabled the use of an algorithm utilising a recurrent deep neural network of the LSTM type as a system for converting the measurement vector into output images. A prototype electrical impedance tomograph was used in the research. The LSTM network, which is often employed for time series classification, was used to tackle the inverse problem. The task of the LSTM network was to convert 448 voltage measurements into spatial images of a selected section of a historical building's brick wall. The 3D tomographic image mesh consisted of 11,297 finite elements. A novelty is using the measurement vector as a single time step sequence consisting of 448 features (channels). Through the appropriate selection of network parameters and the training algorithm, it was possible to obtain an LSTM network that reconstructs images of damp brick walls with high accuracy. Additionally, the reconstruction times are very short.

**Keywords:** electrical tomography; moisture detection; machine learning; neural networks; long short-term memory (LSTM)





## 1. Introduction

A significant concern is the presence of moisture inside brick walls, which can have serious consequences for the health of people who live in these structures and the maintenance and dependability of the buildings in question [1,2]. Monuments and medieval and historical structures are particularly vulnerable to moisture damage [3]. In order to protect the foundations of buildings built more than a century ago from soil moisture, simple procedures were used. Furthermore, as time has passed, the current security measures have deteriorated. Even while periodic improvements might help alleviate the condition, it is common for these projects to be abandoned due to financial constraints. Because of these lapses, historic buildings and structures are being deconstructed, and monuments that may still be attractive places to live or work are being demolished despite their age and age-related decay. The building materials employed in the construction of historical structures are usually invariably permeable in nature. Capillary water is a phenomenon that occurs in ceramic bricks and is quite common. It is the principal method by which moisture travels from the soil up the walls in which capillary leakage is a problem [4]. Moisture in the walls of structures degrades the physical formation of the masonries, resulting in the deterioration of the building's structural integrity. In the worst-case scenario,

this might result in a construction disaster. Therefore, water-soaked load-bearing walls are not only an economic concern that lowers the value of real estate but also a possible hazard to human life and health.

According to scientific evidence, inadequate foundation insulation might result in long-term moisture in the walls. As a result, harmful fungi, bacteria, and mould begin to develop [2,5]. Another negative element affecting the harmful effects of moisture is the presence of chemical compounds in the water composition. For example, carbonates, sulfates, nitrates, and chlorides can be found in the water that seeps into the foundations from the surrounding soil. In addition, the high concentration of salt in the water is particularly detrimental to the walls. It accelerates both the physical degradation of plasters as well as the formation of stains and discolouration, all of which have a negative impact on the look of the facade.

The increased likelihood of various forms of allergies among individuals who are living in the building is one of the health risks associated with excessive moisture inside bricks and walls. It has been discovered that there is a relationship between the amount of moisture present in a building and the prevalence of illnesses such as conjunctivitis, asthma, and rhinitis in people. When considering the difficulties stated above caused by high humidity in building walls, especially those considered very old or historical, the problems mentioned should be regarded as important.

In order to solve the problem of the presence of water in the walls, a prior diagnosis must be performed to identify any wet spots. All testing techniques for moisture in masonry can be split into two categories: invasive (destructive) and non-invasive (non-destructive). The gravimetric (drying-weighing) method of determining the water content in brickwork is thought to be the most accurate [6]. However, it is a disruptive procedure that necessitates the removal of a wall component. The main disadvantage of this method is the requirement to gather samples, which necessitates physical damage (hole drilling) to the wall, which is unacceptable, especially in the case of structures protected as historical monuments.

Indirect approaches are another type of invasive technique. In general, compared with the gravimetric approach, these methods have the advantage of allowing for more minor sampled masonry (carbide and Karl Fisher's methods). However, the benefits do not stop there. The electrical resistive approach does not necessitate the collection of samples, but it requires drilling holes in the wall to insert steel rods. It is quite time-consuming and interferes with the wall substance. Thus, it is not a viable option for historical structures. When it comes to damaging measures, there is always a trade-off.

On the one hand, we attempt to reduce interference by modifying the structure of the tested wall as little as possible, which implies reducing the number of measurement sites. On the other hand, the number of measurement points should be maximised to obtain the most precise measurement result. It is yet another significant disadvantage of this strategy.

The advantage of non-destructive indirect approaches is that they do not require a set number of measuring points. However, on the other hand, these methods have the drawback of only examining the humidity of the walls qualitatively rather than quantitatively. In reality, this implies that they cannot be utilised to make precise wall humidity measurements (e.g., percentages). Instead, they can only determine which sections are humidified and which are not.

Medical computed tomography (CT) is the most well-known. However, as the pace of information technology development has accelerated, industrial tomography, also known as process tomography, has gained prominence. Electrical impedance tomography (EIT) [7], which was utilised in the stated research to determine the presence of moisture in the walls of a historic structure, is a type of electrical tomography. Electrical capacitance tomography (ECT) [8–13] and electrical resistivity tomography (ERT) are other types of electrical tomography. ECT is frequently used to identify air crystals or bubbles within pipes, tanks, or industrial reactors [8,11]. On the other hand, ERT is utilised to recognise geological structures [14–16], such as aquifers, aggregates, and metal deposits [8,13,17]. With ECT, the electrical permittivity is determined [18–20], but with EIT and ERT, the

conductivity is reconstructed [2]. Apart from electrical tomography (EIT, ECT, ERT) [15] and computed tomography (CT) [21], there are numerous other varieties, which can be classified according to the physical phenomenon used: X-rays [22], sound waves [23], magnetism [24–26], electromagnetic waves, and visible light [27].

Due to the physical characteristics of bricks and walls, EIT is the most appropriate tomographic approach for determining moisture content, and so this paper will focus on EIT. The EIT method's primary advantages over other approaches were summarised in [28]. The automation and algorithmisation of tomographic procedures are significant advantages. Apart from the necessary equipment, the algorithm for converting the measured data (inputs) to the output image is critical for obtaining high-quality reconstructive tomographic images. Furthermore, the algorithm's mathematical aim is to solve an inverse problem that is also ill-posed [29]. While mathematical equations are notoriously inefficient in solving inverse and ill-posed problems, machine learning methods with iterative loops parallelised by powerful CPUs and GPUs have enormous potential. Therefore, it is particularly important in the dynamic development of computer techniques, parallel computing, and industrial tomography, especially in the face of changes caused by such phenomena as the fourth industrial revolution (Industry 4.0) [30–32].

Reconstructing the electrical conductivity distribution in the EIT mathematically boils down to solving a nonlinear ill-posed inverse problem using noisy data [33]. All approaches for solving EIT inverse issues can be classified into two categories: deterministic methods and machine learning methods. Deterministic approaches include the Level Set method [34], the Total Variation method [28], the Gauss–Newton method with Tikhonov or Laplace regularisation [2], and others. Machine learning methods include Elastic Net [2,28], Least Absolute Shrinkage and Selection Operator (Lasso) [2], Linear Regression and Least-Angle Regression (LARS) [2], Artificial Neural Networks (ANNs) [2,28], Sparse Bayesian Learning (SBL) [33,35,36], and Logistic Regression [37]. Due to the independence of measurements from time, tomography has not yet used models typical for time series and sequences, including recurrent neural networks.

This research aimed to develop a way to use the LSTM network to solve the inverse problem in electrical tomography. Tomographic data and measurements are not a function of time and therefore are not a natural candidate for processing with predictive models that take sequences or time series into account. On the other hand, neural networks with a feedback loop, especially recurrent deep networks of the long short-term memory (LSTM) type, have more sophisticated and more efficient mechanisms integrated into their structure due to learning and capturing time and sequence dependencies in data than static models. The presented research proves that the mismatch between the tomographic (static) data and the LSTM network can be solved by assuming that the measurement vector is a sequence of values, and each measurement is a single time step.

In other words, we assume that measuring 448 voltages is a process in which the measurements are performed in a specific order. Thus, for all measurement cases, the sequence and time intervals between individual measurements are the same.

This article is divided into five sections. The first section, Introduction, offers a summary of the issues surrounding dampness in buildings, a review of existing methods for determining wall wetness, a description of the types of tomography employed in the stated subject, and the authors' contribution. The second section, titled "Validation Measurements", describes the subject of the research—a historic medieval basilica—and summarises the measurements that were conducted in order to verify the new LSTM-based algorithm. The section "Materials and Methods" considers the technique for gathering measurement data and outlines machine learning models. The fourth section, titled "Results and Discussion", provides research findings from LSTM algorithms and multiple artificial neural network (MANN) experiments. This section includes reconstructions based on real-world data as well as simulated data. Additionally, this part of the manuscript has a discussion that summarises and analyses the collected results. Finally, the fifth section

offers a concise review of the most significant findings from the research and synthesised conclusions. In addition, it contains information on future research proposals.

## 2. Validation Measurements

The research object was the refectory of the Gothic post-Cistercian cathedral complex in Pelplin, Poland (Google Maps coordinates in DMS: N 56°55′40.92″ and E 18°41′53.1″). The Cistercians settled in Pelplin in the 13th century. Already at the end of this century, a brick monastery stood here. In the following centuries, the buildings of the abbey expanded. Many of them were created in the second half of the 15th century when the convent was in a good financial situation. It was then that the vast, gothic basilica was built, the medieval walls of which are filled with elements of furnishings from different eras. Despite numerous reconstructions, the monastery has preserved the medieval spatial layout with a cloister surrounded by cloisters.

In 2014, under the regulation of the President of the Republic of Poland, the post-Cistercian cathedral complex in Pelplin was recognised as a monument to Polish history, which means it was included in the group of monuments under special protection. Since the Gothic basilica in Pelplin is a building with a vibrant past and great historical value, the humidity of its walls cannot be examined by any invasive methods. Figure 1 shows photos of the outer walls of the post-Cistercian cathedral complex in Pelplin.

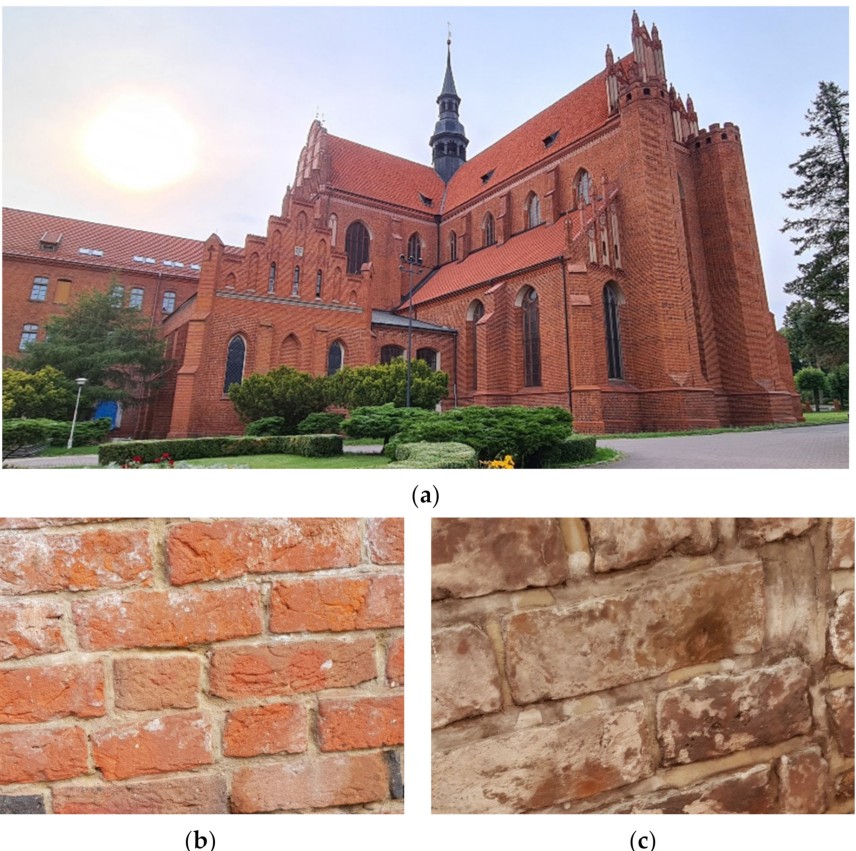

(**a**)

(**b**)                                    (**c**)

**Figure 1.** Part of the post-Cistercian cathedral complex in Pelplin, Poland: (**a**) view from the outside, (**b**) close-up of the external brick wall, (**c**) close-up of the internal brick wall.

The brick wall of the basilica in the Pelplin refectory was chosen as the site of the validation measurements. The electrodes of the electric tomograph were placed against the outer wall from the inside. The brick wall of the refectory of the basilica in Pelplin was subjected to humidity tests to examine the performance of several machine learning algorithms

for imaging tasks. The identical refectory wall was evaluated using a thermal imager and two standard indirect methods to validate the electrical-tomography-based methodologies.

Figure 2 depicts photographs of a part of the tested wall. The FLIR T540 thermal camera was applied (FLIR Systems, Inc., 27700 Southwest Parkway Avenue Wilsonville, OR 97070, USA). A wet surface is cooler than a dry surface due to evaporation. Therefore, moisture settles closer to the floor and along the left vertical corner. It should be noted that infrared photographs only display the surfaces of the inspected items. As a result, it is impossible to determine how the moisture is dispersed inside the wall using infrared photographs.

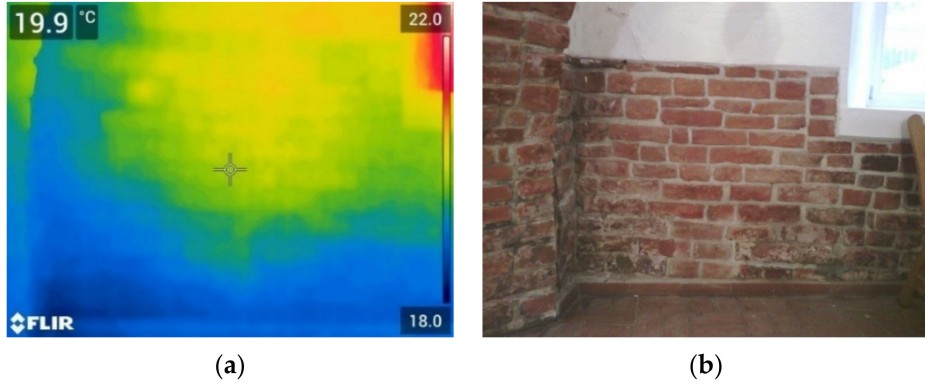

(**a**)  (**b**)

**Figure 2.** The brick wall under consideration: (**a**) thermal image, (**b**) normal image.

The other methods of validation were non-destructive techniques. First, the dielectric approach was used with the Gann UNI-2 (GANN Mess. Regeltechnik GmbH, Stuttgart, Germany) and the B50 ball probe. This meter's test range is roughly 50 mm into the wall. The second way used a Trotec T600 microwave moisture meter (TROTEC GmbH & Co. KG, Heinsberg, Germany) with a measurement range of around 300 mm in the wall. Figure 2a shows the thermal image of the brick wall under consideration.

Figure 3 shows the cross-section of the tested wall. The total width of the wall is 225 cm. The layer of ceramic brick from the inside of the wall is 35 cm thick. The next layer, 145 cm thick, consists of stones. Finally, the outer brick layer is 45 cm thick.

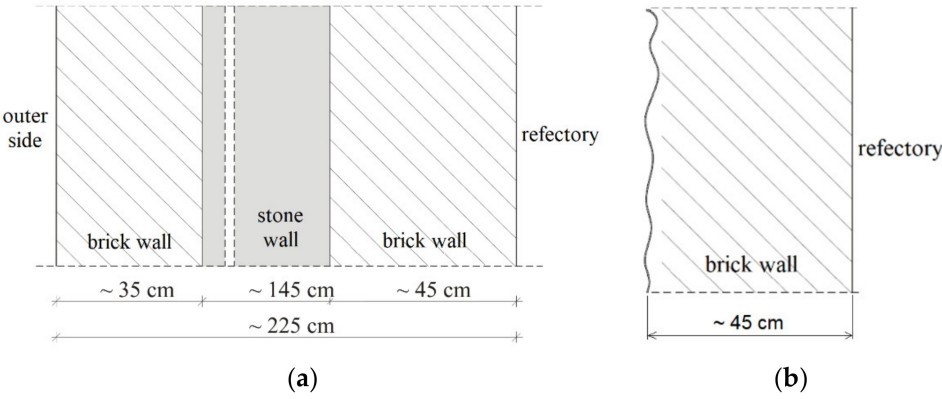

(**a**)  (**b**)

**Figure 3.** Structure of the cross-section of the tested wall: (**a**) the whole wall cross-section, (**b**) the tested piece of a brick wall.

Figure 4 presents the arrangement of all measurement points on the tested section of the wall. All distances in Figure 4 are given in centimetres.

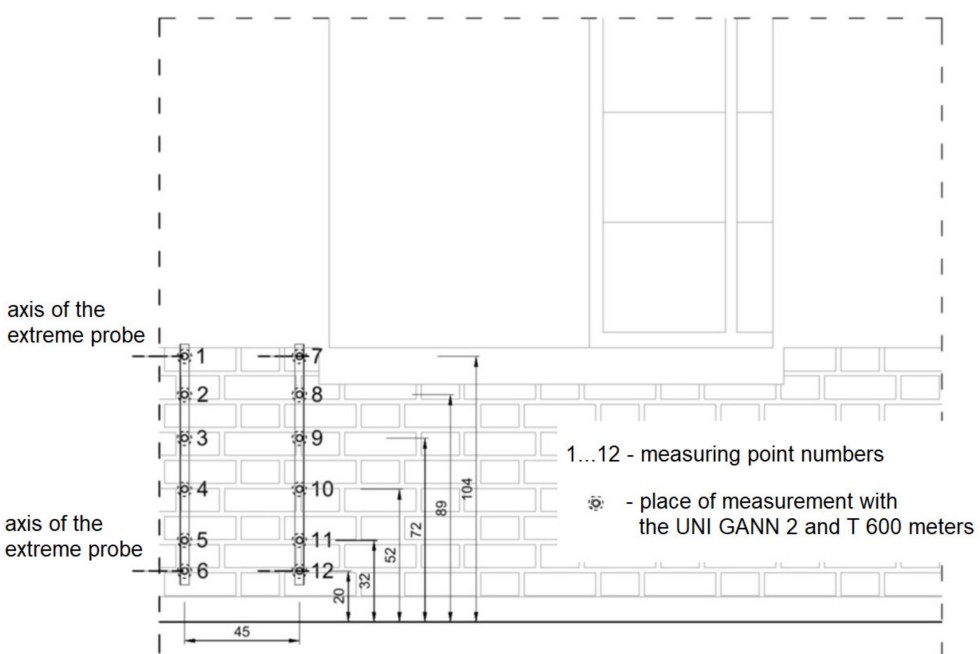

**Figure 4.** Arrangement of the validation measurement points on the surface of the tested wall section. All distances are given in centimetres.

As previously stated, dielectric and microwave techniques are classified as indirect approaches. As a result, these approaches do not generate the mass moisture percentage $U_m$ directly, but rather some unitless quantities $X_D$ and $X_M$. The dielectric technique ($X_D$) is used to determine the change in the dielectric constant of the tested material under the effect of moisture. The microwave ($X_M$) approach, on the other hand, is based on measurements of microwave attenuation as they pass through a moist substance. It means that for both approaches, the mass moisture content $U_m$ is not directly measured during the test but is determined by another physical attribute of the material whose value is proportional to the mass of water contained in it. To determine the mass moisture $U_m$, the correlation between the dimensionless parameter ($X_D$ or $X_M$) measured using a particular method and the mass moisture $U_m$ must be determined.

A correlation link cannot be established for a single tested measurement point. In practice, this correlation connection is established for the entire object, as the minimum number of validation measurements should be approximately 30. Equation (1) illustrates the experimentally determined nonlinear dependencies of the tested wall fragment, which enable the mass moisture $U_m$ to be estimated using dielectric and microwave measurements

$$U_m^D = 0.1667 \cdot e^{0.0335 \cdot X_D} \tag{1}$$

where $U_m^D$ is the mass moisture percentage determined by the dielectric constant $X_D$. Likewise, Equation (2) illustrates the link between the percentage of mass moisture $U_m^M$ and the microwave parameter $X_M$.

$$U_m^M = 0.3425 \cdot X_M - 9.6251 \tag{2}$$

The validation measurements are summarised in Table 1. Column (1) provides the numbers of the measurement points as depicted in Figure 4. Column (2) gives information about the distances of the measurement points from the floor. The findings of dielectric measurements are contained in columns (3) and (4), while microwave measurements are included in columns (5) and (6).

**Table 1.** Results of the validation measurements. Percentage of mass moisture obtained by dielectric $\left(U_m^D\right)$ and microwave $\left(U_m^M\right)$ methods.

| # of Points | The Distance from the Measuring Point to the Floor Level | Dielectric Meter Indication | | Microwave Meter Indication | |
|---|---|---|---|---|---|
| | | $X_D$ | $U_m^D$ (%) | $X_M$ | $U_m^M$ (%) |
| *(1)* | *(2)* | *(3)* | *(4)* | *(5)* | *(6)* |
| 1. | 20 cm | 131.4 | 13.42 | 63.0 | 11.95 |
| 2. | 32 cm | 123.2 | 10.21 | 63.3 | 12.06 |
| 3. | 52 cm | 133.5 | 14.40 | 66.0 | 12.98 |
| 4. | 72 cm | 133.4 | 14.35 | 67.6 | 13.53 |
| 5. | 89 cm | 105.2 | 5.60 | 40.6 | 4.28 |
| 6. | 104 cm | 125.2 | 10.91 | 54.9 | 9.18 |
| 7. | 20 cm | 133.2 | 14.25 | 59.3 | 10.69 |
| 8. | 32 cm | 129.3 | 12.51 | 63.8 | 12.23 |
| 9. | 52 cm | 137.5 | 16.46 | 66.9 | 13.29 |
| 10. | 72 cm | 125.3 | 10.95 | 64.7 | 12.53 |
| 11. | 89 cm | 72.0 | 1.85 | 35.4 | 2.5 |
| 12. | 104 cm | 66.6 | 1.54 | 35.2 | 2.43 |

The validation measures detailed in Table 1 are seen in Figure 5. To ease comparisons between the dielectric and microwave approaches, all data are expressed as mass moisture content $U_m$. The horizontal axes in Figure 5 depict the distances between the measurement points and the ground level. Thick dots denote the exact measurements. Despite changes in $U_m^D$ and $U_m^M$, there is a general trend toward decreased humidity as one moves farther from the ground. In Figure 5b, the dynamics of the decline in mass moisture are more pronounced. Although the reduction in humidity in Figure 5a is less dynamic, it may result from an unequal moisture distribution within the tested wall piece.

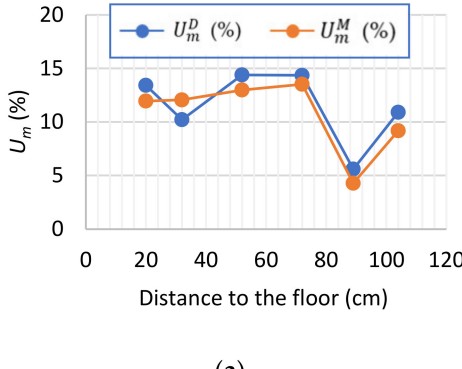

(**a**)

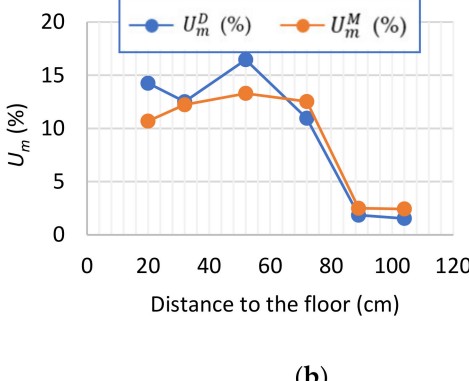

(**b**)

**Figure 5.** Validation of the tested wall moisture by dielectric and microwave methods: (**a**) points 1–6, (**b**) points 7–12.

## 3. Materials and Methods

In order to verify the effectiveness of the concept of using the LSTM network in static measurements of electrical tomography, two predictive models were trained: LSTM and ANN. Then, the results of tomographic imaging obtained with the two methods mentioned above were compared.

The ANN model took into account the approach of creating a tomographic image using a multilayer perceptron using the pixel-by-pixel method. This technique, called multiple neural networks, was described in [38]. Training multiple separate shallow neural networks for individual pixels (with a single output) increases the effectiveness of prediction compared with training multi-output networks. However, it takes place at the expense of extending the training time and, what is worse, the prediction time.

The model consisting of a single LSTM network does not have this disadvantage. The morphology of the LSTM network predestines it to process large amounts of data. As a

result, training a single LSTM network with 448 inputs and 11,297 outputs is not too much of a challenge. Figure 6 shows a comparison of multiple ANNs and homogeneous LSTM.

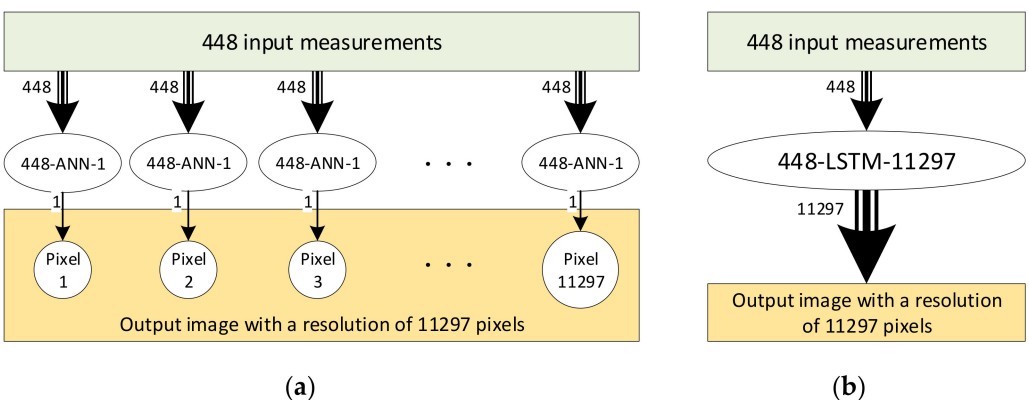

**Figure 6.** Methods for converting measurements into images used in the study: (**a**) pixel-by-pixel with multiple ANNs, (**b**) single LSTM.

The tomograph prototype with the electrodes was entirely designed and manufactured in the laboratory of Nertix SA. The "hybrid wall moisture measurement system" device is intended to monitor the distribution of moisture in the walls of historic buildings without the need to make test boreholes. However, due to the ubiquity of the measurement method, it can also be utilised for objects with differing electrical properties. As the name suggests, the hybrid tomograph device is designed to merge two measuring methods: the EIT impedance and the ECT capacitance measurement [39–41]. The first versions of the hybrid tomograph built in our laboratory were based on microprocessor systems. Measurements were performed sequentially on 8/16/32 channels, and the sampling rate oscillated around 100 kilo Samples Per Second (kSps). Sequentially, the capacitive measurement was performed independently (sequentially) from the impedance method, resulting in a significant measurement time extension. In the later stages of the project's development, there was a need to create a faster version while ensuring the correctness of measurements, which resulted in the need to create version 2.0.

The hybrid tomograph in version 2.0 (next generation), unlike its predecessors, was based on a set of Intel Altera Cyclone IV and Cyclone V FPGA systems. It allowed for the use of parallel function blocks independent of each of the channels. The measurement roles have been divided into eight Cyclone IV systems, one each for four measurement inputs, which, coupled with the ADS8588 multi-channel A/D converter and signal exceeding 0 V detection circuits, perform measurement functions. The signal thus measured is then partially filtered out using the FIR2 filter. Then, the RMS voltage value is calculated, and the signal's phase shift between the excitation current and the voltage. The measured values are sent to the control unit, which transfers the data via Ethernet, stores it in mass memory, and/or performs reconstruction in real-time.

The main limitation regarding the measurement speed in the case of the 2.0 hybrid is the signal period that we use to excite the object. In the case of a 1 kHz signal, it is 1 ms multiplied by the number of electrodes, which in the case of 32 electrodes allows for a theoretical measurement speed of 32 ms. Really, due to the need to stabilise the current after switching the excitation, we use one dead period, which still allows for a sampling rate below 100 ms. As the frequency of the excitation signal increases, the measurement speed increases proportionally. Figure 7 presents the EIT measurement station located next to the brick wall of the refectory of the basilica in Pelplin. There is often a problem with installing electrodes on the tested object (Figure 7b). Often, the tested object has an uneven surface to which it is difficult to attach the electrodes correctly. To address these issues, a flexible electrode mounting system was developed to ensure correct contact between the electrode and the object under test and to accommodate surface irregularities.

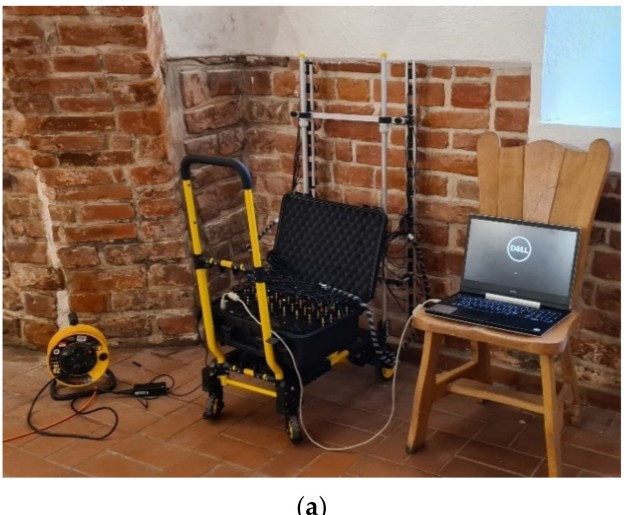
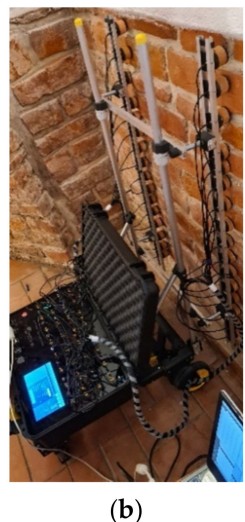
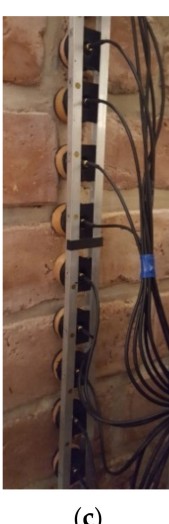

(**a**)   (**b**)   (**c**)

**Figure 7.** The brick wall of the refectory of the basilica in Pelplin: (**a**) EIT measurement station, (**b**,**c**) the electrodes applied to the wall.

The electrode system is made in the form of a flexible body, which consists of: an articulated handle with a flexible joint, in which the connection socket is mounted, and a shock-absorbing sponge. Elements of the electrode body were made in 3D technology from plastic. Two sleeves connect them in place, one inside the other, and they are prevented from separating by a flange located on the upper sleeve. To ensure the body's flexibility, it has been equipped with a rubber ring placed between the sleeves. This connection allows the elements to move relative to each other along the axis. An additional element improving the system's flexibility is a sponge made of polyurethane foam that connects the body with the measuring electrode. Thanks to this solution, the freedom of movement of the electrode are significantly extended, and the pressure is improved.

Electrodes are used to conduct a series of currents through the object being studied, and the voltages they produce are measured using the same electrodes. The first projection (1) and the second projection (2) show the opposite approach of gathering boundary potential data for a cylindrical volumetric guide and 16 equally spaced electrodes. However, when electrodes are put along metal strips such as the one depicted in Figure 8, they must be organised in a straight line rather than a circular pattern as depicted in the figure. A detailed description of the measurement process can be found in [2].

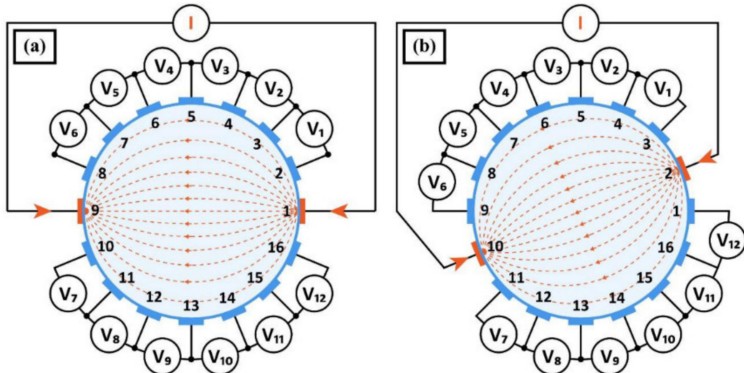

**Figure 8.** Opposite measurement method in a system of 16 electrodes: (**a**) first measuring cycle, (**b**) next measurement cycle [2].

### 3.1. Training Data Preparation

The data set generated by the simulation was utilised for training the predictive models. A total of 30,000 simulation cases were developed to represent empirical observations. The measurement data vector had 448 values, which correspond to the voltage drops between the individual electrode pairs. The spatial output image was segmented into 11,297 pixels. A pixel is a single finite element placed on a mesh. The finite element method and Eidors toolbox for Matlab were used to construct the simulation algorithm [42–44]. Although the problem in the proposed research is binary, it is solved using regression models. As previously stated, the EIT model is binary because it contrasts moist and dry areas. If we establish that the dry surface is a wall with an electrical conductivity of 1 ($\sigma = 1$), and the wet surface has an electrical conductivity of 10 ($\sigma = 10$), we can convert the regression results for binary classifiers using the formula $f(u) = \sigma$ generating real numbers as the output, where $u$ is the measurement vector. The output file will then contain only two classes: 1 and 10 ($\sigma \in \{1; 10\}$). It has been proved experimentally that the binary technique produces unsatisfactory visual results. By maintaining the initial conductance values as real numbers, we may observe two states (dry or wet) and far more intriguing visuals due to the colour difference of individual pixels. Thus, the output values are numerical in any unit that is proportional to conductivity.

### 3.2. The Long Short-Term Memory (LSTM) Neural Network

The architecture of the LSTM network layer used in the described research is presented in Figure 9a. The red line surrounds the LSTM section, which was adapted as a single (first and only) time step. A time series having $C$ features (channels) of length $S$ is denoted by $X$. The binary output including both the latent state and the cell state at time step $t$ is indicated by the symbols $h_t$ and $c_t$ in the diagram. Features ($x_1, x_2, ..., x_C$), in this case, denote the measurement vector consisting of $C = 448$ measurements. The length of the time series $S = 1$. In each LSTM layer, $D = 128$ hidden units were used. The measurement vector was treated as a sequence, i.e., an ordered set of values derived from 448 individual measurements performed at regular intervals. Figure 9b shows the inside of the LSTM block.

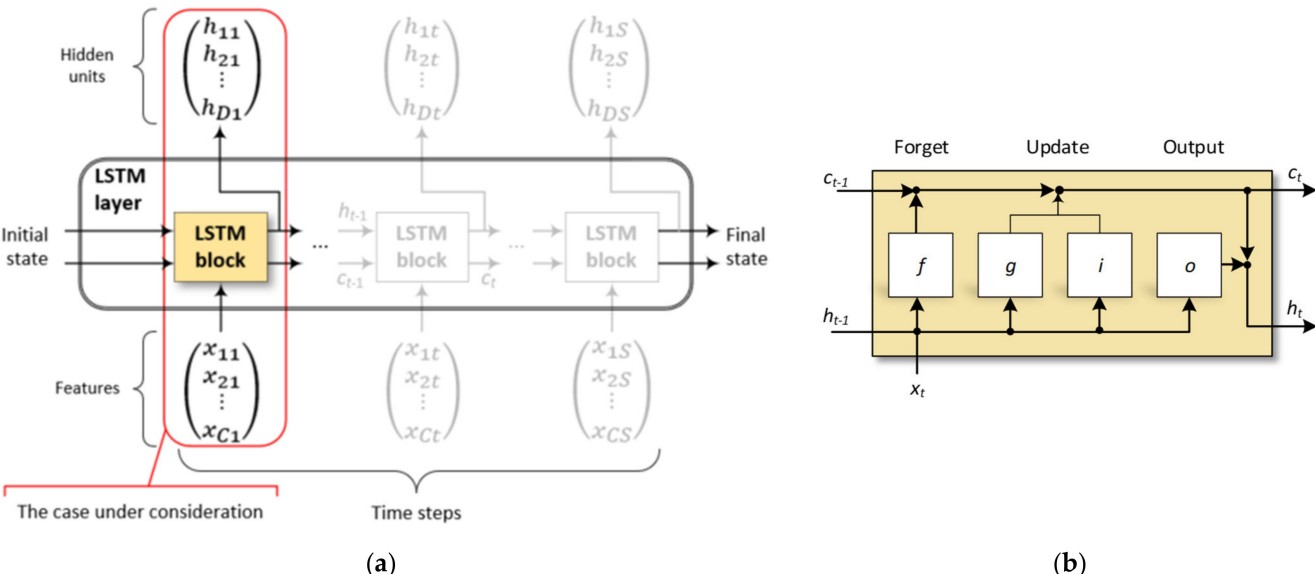

(a) (b)

**Figure 9.** LSTM network structure: (**a**) considered modification of the LSTM network with a single time step, (**b**) LSTM block interior.

The hidden state (alternatively called the output state) and the cell state are the two states that comprise the LSTM layer's state. For example, the layer's output for time step $t$ is hidden. At each time step, the cell state contains information gleaned from previous time

steps. Thus, depending on the input vector, the layer adds or subtracts information from the cell state at each step. The layer is responsible for managing these changes via gates. The following gates manage the layer's cell state and hidden state: input gate (i) controls the cell state update level, forgetting gate (f) controls the cell state reset level, cell candidate (g) adds information to the cell state, and the output gate (o) controls the cell state level to be added to the hidden state.

Figure 9b depicts the data flow at time step $t$. The graphic illustrates how the gates remember, update, and derive cell and hidden states, as well as their interaction. The research employed a six-layer LSTM network. There are no deterministic criteria for selecting network parameters (e.g., number of layers, number of hidden units, normalisation layers, initial weights, or activation functions) for specific types of problems in the case of neural networks. As a result, network and training parameters are empirically chosen. It was also the case here. Formulas (3) can be used to characterise the weights $W$, the recurrent weights $R$, and the biases $b$ [45].

$$W = \begin{bmatrix} W_i \\ W_f \\ W_g \\ W_o \end{bmatrix}, \quad R = \begin{bmatrix} R_i \\ R_f \\ R_g \\ R_o \end{bmatrix}, \quad b = \begin{bmatrix} b_i \\ b_f \\ b_g \\ b_o \end{bmatrix}, \tag{3}$$

The symbols $i, f, o$, and $g$ signify the input, forget, and output gates, as well as the cell candidate, respectively. The state of a cell at a particular time step $t$ is denoted as $c_t = f_t \odot c_{t-1} + i_t \odot g_t$, where $\odot$ represents vector element-wise multiplication. At time step $t$, the hidden state is defined as $h_t = o_t \odot \sigma_c(c_t)$, where $\sigma_c$ is the state activation function. The Equation (4) define the LSTM layer's components at time step $t$

$$\begin{cases} i_t = \sigma_g(W_i x_t + R_i h_{t-1} + b_i), \\ f_t = \sigma_g\left(W_f x_t + R_f h_{t-1} + b_f\right), \\ g_t = \sigma_c\left(W_g x_t + R_g h_{t-1} + b_g\right), \\ o_t = \sigma_g(W_o x_t + R_o h_{t-1} + b_o), \end{cases} \tag{4}$$

where $\sigma_g$ denotes the gate activation function. The sigmoidal activation function was employed in the LSTM network layers. It is defined as $\sigma(x) = (1 + e^{-x})^{-1}$. Table 2 presents the parameters of the LSTM neural network used to solve the tomographic inverse problem. The LSTM model contains two bidirectional LSTM (BiLSTM) layers with 128 hidden units each. Experiments have demonstrated that fewer hidden units degrade the network's quality, but increasing the number of hidden units and adding new layers prolongs the learning process but does not improve the LSTM network's efficacy. Sequence Input is the initial layer in the LSTM model. The sequence input layer is used to feed the network the measurement data vector. Another instance is BiLSTM's bi-directional layer. The bidirectional LSTM layer learns long-term correlations between signal time steps or sequence data (feedback) in both directions. These interactions are critical when the network needs to learn at each time step from the full time series. The third layer is batch normalisation, which is used to avoid overfitting the model. BiLSTM's fourth layer is identical to the second. The model's fifth layer is fully connected. The weight matrix is multiplied by the numerical input values in this layer, and the bias vector is also added. The final, sixth regression layer generates the tomographic image's 11,297 pixel values.

**Table 2.** Layers of the LSTM neural network.

| # | Layer Description | Activations | Learnable Parameters (Weights and Biases) | Total Learnables | States |
|---|---|---|---|---|---|
| 1 | Sequence input with 448 dimensions | 448 | – | 0 | – |
| 2 | BiLSTM with 128 hidden units | 256 | Input weights: $1024 \times 448$; Recurrent Weights: $1024 \times 128$; Bias: $1024 \times 1$ | 590,848 | Hidden state: $256 \times 1$ Cell state: $256 \times 1$ |
| 3 | Batch normalization with 256 channels | 256 | Offset: $256 \times 1$ Scale: $256 \times 1$ | 512 | – |
| 4 | BiLSTM with 128 hidden units | 256 | Input weights: $1024 \times 256$; Recurrent Weights: $1024 \times 128$; Bias: $1024 \times 1$ | 394,240 | Hidden state: $256 \times 1$ Cell state: $256 \times 1$ |
| 5 | Fully connected layer | 11,297 | Weights: $11,297 \times 256$; Bias: $11,297 \times 1$ | 2,903,329 | – |
| 6 | Regression output | 11,297 | – | 0 | – |

Classifier options were specified before beginning the training process. The number of epochs was set to 500. The size of the minibatch was set to 500, too. The number of iterations per epoch was 64. The training was conducted at a constant rate of 0.001. This property was established to make the learning process more accurate. The gradient threshold was set to 1 to stabilise the training process by preventing the gradient from becoming too big. We employed an adaptive moment estimation optimiser (ADAM). The ADAM is more effective with recurrent neural networks (RNNs), including LSTM, than stochastic gradient descent with momentum (SGDM). A regression layer is the last one in the considered model. The half means that the regression layer calculates the squared error loss for regression tasks. A regression layer must come after the final fully connected layer in common regression problems. The rooted mean squared error is calculated as follows for a single observation according to Formula (5)

$$RMSE = \sqrt{\frac{\sum_{i=1}^{N}(y_i - \hat{y}_i)^2}{N}}, \tag{5}$$

where $N$ is the number of responses, $y_i$ denotes the pattern output, and $\hat{y}_i$ denotes the LSTM prediction for the response $i$ [46]. For sequence-to-one regression networks, the loss function of the regression layer is the half-mean-squared error of the predicted responses, not normalised by $N$. Formula (6) satisfies the loss function used.

$$Loss = \frac{1}{2}\sum_{i=1}^{N}(y_i - \hat{y}_i)^2, \tag{6}$$

During training, the software calculates the mean *Loss* over the observations in the mini-batch. The training status of the LSTM network is depicted in Figure 10.

The training progress graph shows the RMSE error of the learning process. It reflects the regression accuracy of each mini-batch. For perfect training progress, the RMSE tends to zero. At the end of the learning process, the RMSE value hovers around 80. This value is influenced by non-normalised outputs, which can reach values from 1 to 10. The training took about 8 h. The training process was completed after 535,000 iterations. The reconstruction time for a single observation was 0.010577 s. The calculations were performed on the Intel® Core ™ i5-8400 2.80GHz processor, 16GB of RAM, and an NVIDIA GeForce RTX 2070 GPU. The GPU was used.

Figure 10b shows the LSTM learning process based on the loss rate. The graph shows the training loss for each mini-game. When training is perfect, the loss should drop to zero. The shape of this graph confirms all the information that results from Figure 10a.

Lower values of the Loss function from the RMSE result from the calculation differences in Formulas (5) and (6).

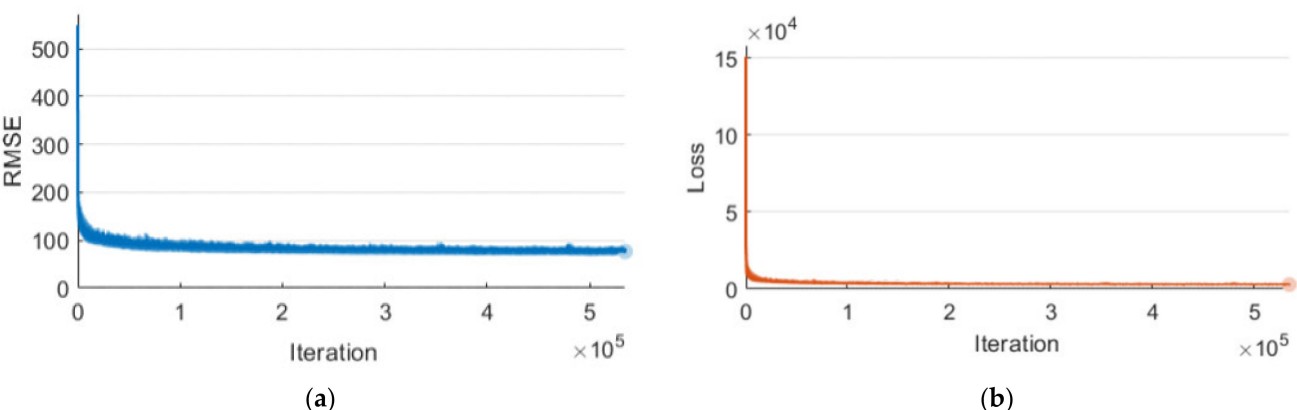

**Figure 10.** Training progress of the LSTM: (**a**) graph of *RMSE*, (**b**) graph of *Loss*.

### 3.3. The Multiple Artificial Neural Network (MANN)

The previously stated pixel-by-pixel model structure with multiple ANNs (MANN) has two layers with transfer functions: a hidden layer and an output layer. The MANN structure can be represented as a $(448 \rightarrow 10 \rightarrow 1) \times 11,297$ system that trains 11,297 distinct, single-output prediction models that each determines the value of one pixel in the tomographic image. All neurons use the hyperbolic tangent sigmoid transfer function in the hidden layer to compute weights. The output layer employs the linear transfer function. Because its purpose is to provide basic information that transfer functions can utilise to construct weights and bias values during the ANN training process, the training set covers up to 70% of all cases. The cases in the testing and validation sets do not immediately participate in the training process. The goal of the validation set in the ANN training technique was to prevent overfitting and give the network generalisation capabilities. The network training algorithm calculated a validation set error using an early stopping strategy [47]. If the validation error continues to rise over the following six epochs, the network training process is terminated. The training process shown in Figure 10 was performed on a single, randomly chosen pixel.

Figure 11a depicts the MSE values calculated for each training, validation, and testing set iteration. The best validation results were achieved with MSE = 4.1179. As illustrated, two lines (validation and test) are very consecutive and have the shape of a regular hyperbola, indicating that the ANN is not overfitted. Only the training error is significantly different from the validation and test errors. To prevent overfitting, the training process ended automatically after the 15th epoch, while the parameters for the trained network were adopted six epochs earlier—for the ninth epoch.

Figure 11b depicts the gradient values for specific validation set iterations as a function of the number of successive MSE increases for that set. It is significant for the anti-overfitting strategy (early stopping) because the ANN learning process is interrupted after six consecutive MSE validation errors increase. The graph also shows the evolution of the momentum (mu) value during the 15 training epochs.

Figure 11c depicts the MSE histogram for several sorts of sets. It demonstrates that the vast majority of deviations have modest values close to zero. Thus, it is an excellent indicator of the neural network's quality. Furthermore, as can be observed, the MSE histogram has the shape of a Gaussian curve, indicating that the ANN training was effective.

Bins are the number of vertical bars (see the graph). The total error from the neural network ranges from $-9.4245$ (the leftmost bin) to 8.9145 (the rightmost bin). This error range is divided into 20 smaller bins, so each bin has a width of $\frac{(8.9145-(-,9.4245))}{20} = 0.91695$. Each vertical bar represents the number of samples from the dataset that lie in a particular

bin. For example, at the midpoint of the histogram, you have a bin corresponding to an error of 0.2034, and the height of that bin for the validation dataset is about 300. This means that about 300 samples from the validation dataset have an error that lies in the following range: $\left(0.2034 - \frac{0.91695}{2},\ 0.2034 + \frac{0.91695}{2}\right)$, so finally the width of this bin is $(-0.2551, 0.6619)$. The MANN training process took approximately 43 h and the reconstruction time was about 17 s.

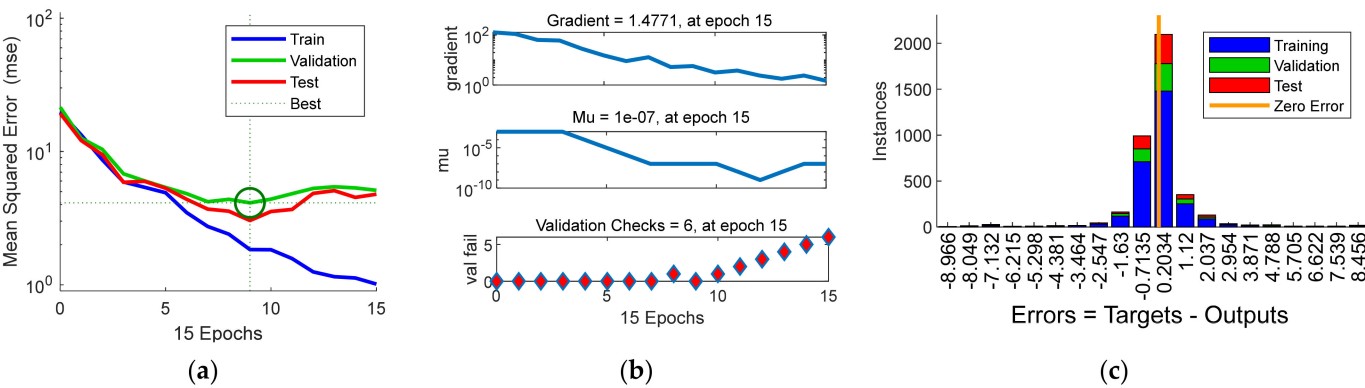

**Figure 11.** The MANN training process for the arbitrary pixel: (**a**) best validation performance plot, (**b**) early stopping strategy, (**c**) error histogram with 20 bins.

## 4. Results and Discussion

To thoroughly validate the LSTM approach, comparisons of EIT reconstructions based on real and simulated measurements were performed. The brick wall part was rebuilt utilising two methods: the novel LSTM approach and the MANN method (as a high-performance reference method) described in [38]. The reconstructions of the real measurements only allow for a subjective judgement based on the overall impression. We must rely on point validation measurements because we do not know the real moisture distribution inside the tested wall piece. Spot measurements are a guideline that can be used to forecast a reconstruction image very roughly. The first part of this section is devoted to the examination of reconstructions based on real-world data. Using simulation data is a much better, more objective technique to judge the quality of the reconstruction. One of their advantages is determining the divergence between the reference value of individual pixels and the value reconstructed by the model. Furthermore, quantitative criteria can be used to assess the quality of images reconstructed from simulation data. Such analyses were performed in the second part of this section.

### 4.1. Visualisations of Real Measurements

Figure 12 shows reconstruction images based on EIT measurements in the refectory of the basilica in Pelplin. EIT measurements were made on the test stand described in Section 2 Validation Measurements. The same measurement vector was processed by two methods: LSTM and MANN. All pictures show the tested fragment of the wall with dimensions of $100 \times 65 \times 50$ cm (height, width, depth). The two vertical rows of black dots in the images mark where the wall meets the electrodes. The electrodes were placed on two metal bars with 16 electrodes on each of them. The images reconstructed using the LSTM method are placed in the upper row and reconstructed by the MANN method in the lower row. Pictures (a) and (d) show the front view and therefore give the impression of being flat (2D). Images (b) and (e) visualise the examined wall section from the outside in a semi-side view (3D). Images (c) and (f) show a spatial view from the point of view of an observer inside the examined wall segment.

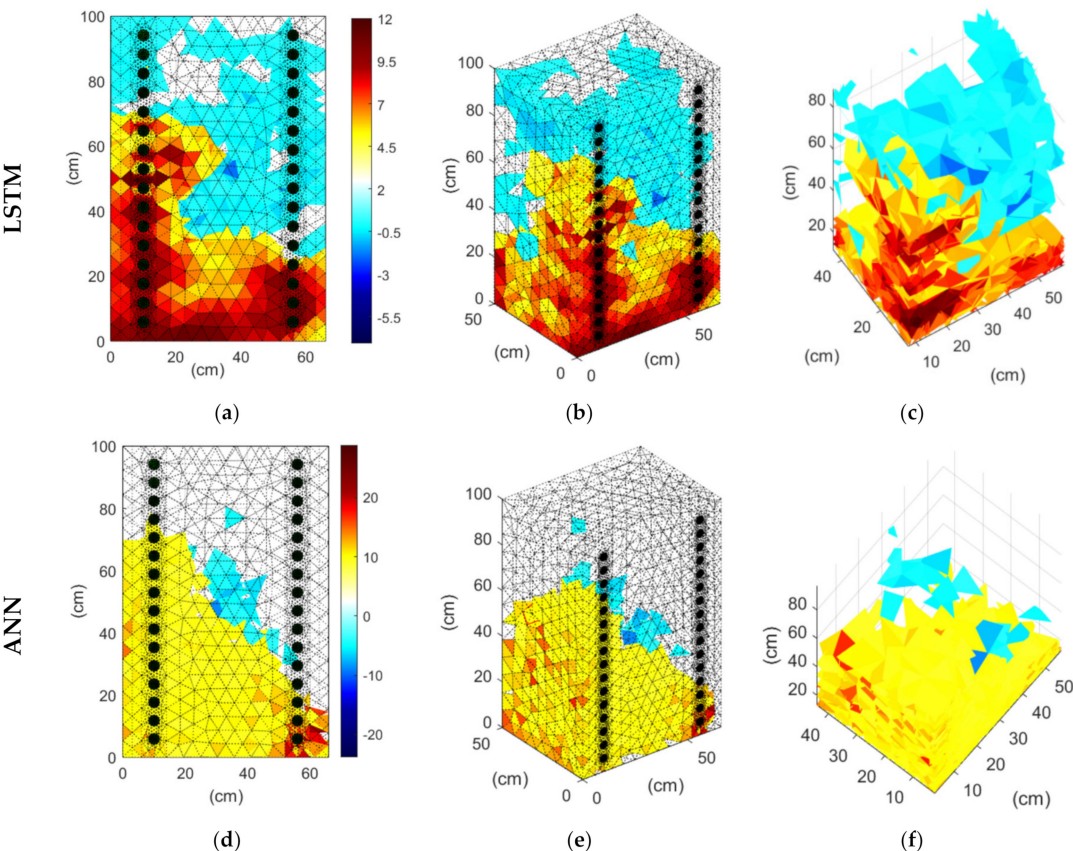

**Figure 12.** Reconstructions obtained from real measurements. Images (**a**–**c**) were obtained with LSTM, where (**a**) is the front view, (**b**) is an exterior spatial view, and (**c**) is a spatial view from an internal perspective. Images (**d**–**f**) were created with the MANN algorithm and are analogous to (**a**–**c**).

Please keep in mind that the colours do not correspond to precise moisture levels. Individual pixels are coloured according to their conductivity, which is essential the more humid the object being examined. This is demonstrated by the coloured bars, where greater values indicate more humidity. As previously stated, no reference image exists for reconstructions based on real data. However, the LSTM images appear to be more homogenous and unambiguous than the MANN reconstructions.

### 4.2. Assessment of the Reconstructions Based on Simulation Data

Four commonly used measures were used to assess the quality of tomographic reconstructions objectively: root mean square error (RMSE), normalised mean square error (NMSE), relative image error (RIE), and image correlation coefficient (ICC). The root mean square error is calculated according to Formula (5). The NMSE is calculated according to (9)

$$\text{NMSE} = \frac{||\boldsymbol{y} - \hat{\boldsymbol{y}}^2||}{||\boldsymbol{y} - \overline{\boldsymbol{y}}^2||} \ , \tag{7}$$

where $\boldsymbol{y}$ is the ground-truth (reference) conductivity distribution, $\overline{\boldsymbol{y}}$ denotes the mean reference ground-truth conductivity distribution, $\hat{\boldsymbol{y}}$ is the reconstructed conductivity distribution, and $||\cdot||$ indicates the L2–norm vector [48]. Therefore, the RIE is calculated as Function (8).

$$\text{RIE} = \frac{||\boldsymbol{y} - \hat{\boldsymbol{y}}||}{||\boldsymbol{y}||} \tag{8}$$

The ICC is described by Equation (11)

$$\text{ICC} = \frac{\sum_{i=1}^{n}(y_i - \overline{y})(\hat{y}_i - \overline{\hat{y}})}{\sqrt{\sum_{i=1}^{n}(y_i - \overline{y})^2 \sum_{i=1}^{n}(\hat{y}_i - \overline{\hat{y}})^2}} \ , \tag{9}$$

where $\overline{\hat{y}}$ denotes the mean EIT reconstruction conductivity distribution. The lower the RMSE, NMSE, and RIE, and the greater the ICC, the higher the tomographic image quality. Thus, ICC = 1 indicates perfect reconstruction, whereas ICC = 0 indicates the worst-case scenario.

Figure 13 shows four cases of different simulation-generated moisture. The images in the first column (a, d, g, j) act as patterns. The images (b, e, h, k) were reconstructed by the LSTM method, while the images in the third column (c, f, i, l) were generated by the MANN method. Visual and subjective comparative assessments favour the LSTM method. It is especially clearly visible in the examples of cases 3 and 4. In these cases, the mapping of the moisture shapes obtained by the MANN method is worse. It is also worth noting that the reconstructed images' colour calibration is different from that of the reference images. It should be realised that the purpose of tomography is not a presidential, quantitative measurement of humidity but the visualisation of areas with increased humidity. Increased humidity is an imprecise term and must be related to the background humidity. On the other hand, determining the background humidity should enable the best visualisation of areas slower than the background. Therefore, colour calibration is an important element in the process of reconstructing tomographic images.

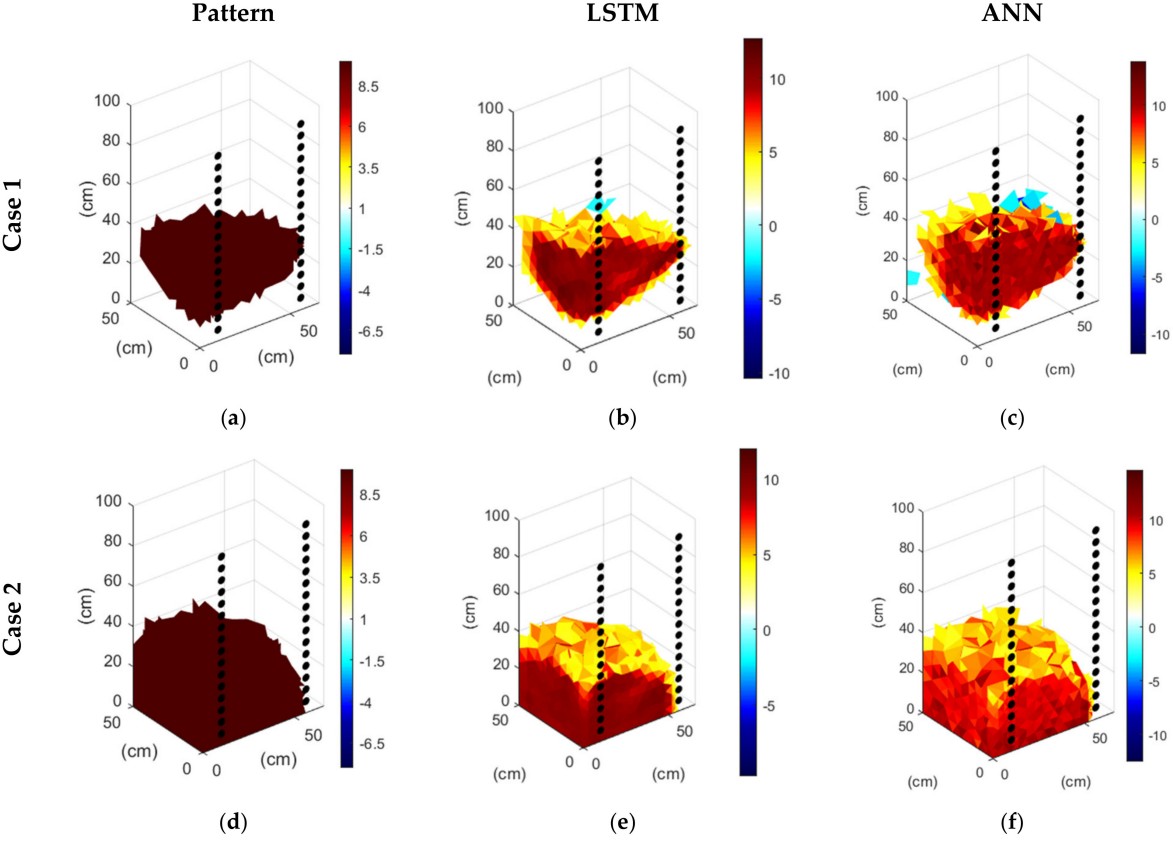

**Figure 13.** *Cont.*

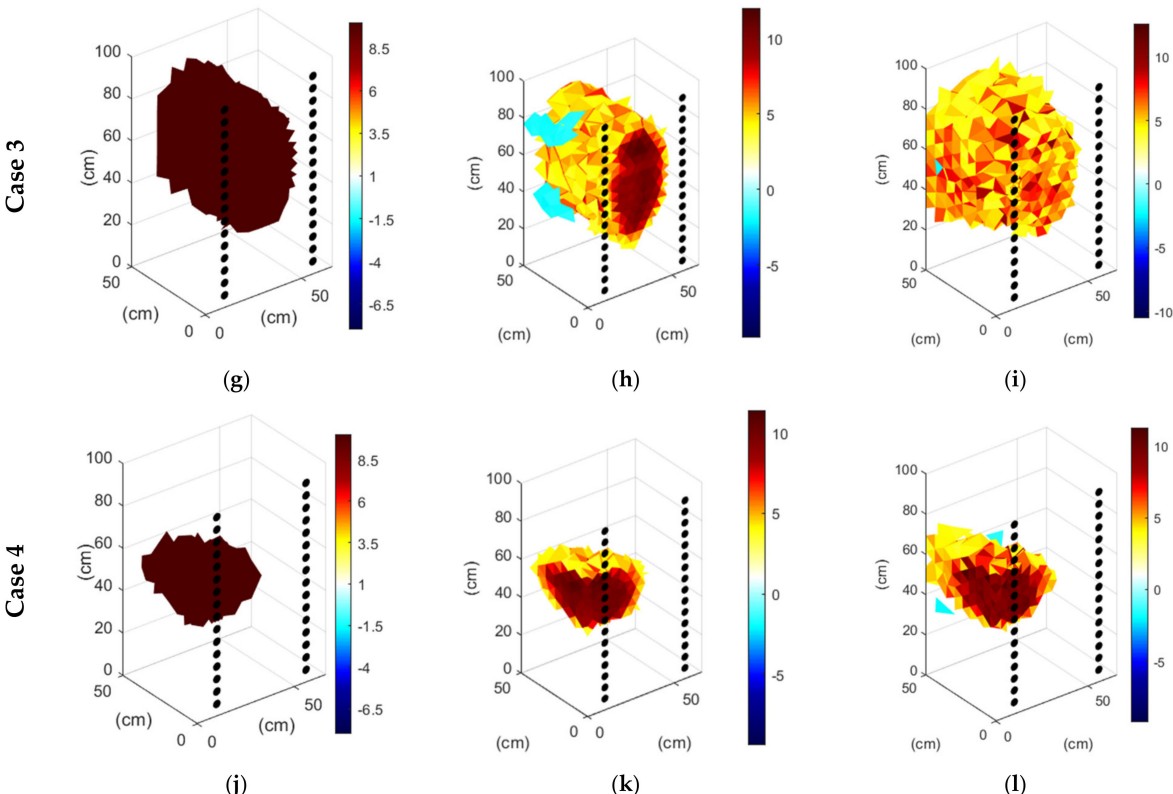

**Figure 13.** Reconstructions obtained from simulation measurements: (**a,d,g,j**) are reference images, (**b,e,h,k**) are images obtained using LSTM, (**c,f,i,l**) are images obtained using ANN.

Table 3 shows a comparison of the LSTM and MANN methods. For this purpose, four criteria defined as indicators were used: RMSE, NMSE, RIE, and ICC. Four different cases of moisture content were assessed. The last four lines of the table contain information about the method that gives the best result for a given case in the context of each of the measures. As can be seen, in all cases, better results were obtained using the LSTM method. It should be objectively stated that some of the differences are very small. For example, a comparison of case 1 almost indicates a draw, but case 3 already brings significant differences. When comparing the two methods, one more factor should be considered, namely the reconstruction time. The model's ability to generate images in a short time is of great importance, especially in the case of measurements performed during dynamic industrial processes. Both the MANN and LSTM models were trained on the same personal computer. The LSTM reconstruction time was 0.010577 s, and the MANN reconstruction time was 17.096585 s. In practice, the difference is 1616 times in favour of the LSTM method.

**Table 3.** Reconstruction quality indicators for homogenous methods—testing set.

| The Method | Indicator | Methods of Reconstruction | | | |
|---|---|---|---|---|---|
| | | Case 1 | Case 2 | Case 3 | Case 4 |
| LSTM | RMSE | 1.419 | 0.812 | 1.445 | 0.548 |
| | NMSE | 0.190 | 0.076 | 0.179 | 0.070 |
| | RIE | 0.259 | 0.178 | 0.321 | 0.219 |
| | ICC | 0.945 | 0.972 | 0.918 | 0.963 |
| ANN | RMSE | 1.436 | 1.040 | 3.332 | 0.690 |
| | NMSE | 0.192 | 0.097 | 0.414 | 0.089 |
| | RIE | 0.260 | 0.201 | 0.488 | 0.246 |
| | ICC | 0.943 | 0.966 | 0.812 | 0.954 |
| Winning method: | | LSTM | LSTM | LSTM | LSTM |
| | | LSTM | LSTM | LSTM | LSTM |
| | | LSTM | LSTM | LSTM | LSTM |
| | | LSTM | LSTM | LSTM | LSTM |



## 5. Conclusions

This manuscript presents the original algorithmic concept of applying a recurrent deep LSTM network to the static problem of tomographic image reconstruction. The concept was applied to imaging moisture inside a brick wall using electrical impedance tomography. By treating the measurement vector as a single time step sequence signal, the LSTM network was successfully trained. The high quality of the reconstruction was confirmed by comparing the images generated by another high-efficiency method, MANN. The presented method enables spatial visualisation of the moisture distribution inside the wall. It is a fundamental difference compared with classic indirect methods, which only test the humidity at selected wall points. A significant advantage of the LSTM method is the very high reconstruction speed, which opens up new application possibilities, especially in the area of automated, dynamic industrial processes. The great advantage of the LSTM model's speed over the MANN model results from the many times lower complexity of the LSTM model. The MANN model consists of 11,297 individually trained ANN models operating in parallel or in series. Individual reconstruction of an image requires running 11,297 ANN processes. This is the reason for the long delay. With the LSTM model, this issue does not occur. A single LSTM model, although more complex than any of the individual ANN models, reconstructs images much faster than several thousand ANN models. Future research will be directed towards sequencing the moisture expansion processes inside porous materials. More sophisticated modifications to the input vector (data preprocessing) are planned to exploit the LSTM network's potential fully.

**Author Contributions:** Development of the system concept and writing—original draft preparation, G.K.; Hardware, measurement methodology, image reconstruction and supervision, and writing—review and editing, T.R.; Development of the concept of measurements in a historical building, preparation of measurement stations, measurements, and development of measurement methodology, A.H.; Literature review, formal analysis, general review and editing of the manuscript, and funding acquisition, Ł.S.; Development of the numerical methods and techniques, T.W.; Data curation and preparation of descriptions in the manuscript, M.K. All authors have read and agreed to the published version of the manuscript.

**Funding:** This research received no external funding.

**Institutional Review Board Statement:** Not applicable.

**Informed Consent Statement:** Not applicable.

**Data Availability Statement:** Not applicable.

**Conflicts of Interest:** The authors declare no conflict of interest.

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
