# Peer review of "The Concept of Using LSTM to Detect Moisture in Brick Walls by Means of Electrical Impedance Tomography"

_energies, doi:10.3390/en14227617_

Round 1

Reviewer 1 Report

Main comments

  1. The authors do not discuss the physical basis of their research at all. I could not find frequency range of their impedance spectrum. I do not know what they are measuring as the unitless (dimensionless?) quantity XD. If I measure the impedance, I have it in Ohms. What is this XD? If this is the effective permittivity, then how can the authors extract this parameter from the impedance spectra? I read in the manuscript about the application of microwave technique and another dimensionless parameter XM. However, in Figure 3 I only see wires suitable for RF. Where are the waveguides, emitters and receivers for transporting microwave energy located? I do not know between which electrodes the authors measure impedance. What is the construction of electrodes? What is the quality of electrode-wall contact? By the way, in Figure 3 I see 2*16 electrodes, while there are 2*6 electrodes in Figure 6. Is the 2*6 electrodes configuration only used for the mass humidity validation? What is parameter Um: mass humidity, mass moisture, mass moisture percentage? What are its units? I cannot see linear dependence in eq. (1). How can the parameter XD be measured at single point, if it is analogous to impedance? Again, what is XD: impedance, conductivity, permittivity?
  1. Other comments refer to the mathematical part. According the training, the authors run 30,000 simulations to represent empirical observations. What was the mathematical model for these simulations? I think it was necessary to use the permittivity and conductivity of brick and stone walls, cement mortar and the dependence of these parameters on humidity. What was the authors’ model? What was the strategy for these simulations? Do the authors use a random distribution of moisture in the walls, or do they preliminarily solve a filtration problem to find a distribution that has physical meaning? The results of reconstruction excite serious doubt. I cannot see in Figures 11 and 12 the boundary between brick and stone walls. It looks as the wall is homogeneous.

I believe the manuscript should be rejected for serious revision.

Reviewer 2 Report

The present work attempts to introduce the LSTM for moisture detection using impedance tomography. Considerable and mandatory revision is required to improve the quality of the manuscript. Please find below the detailed queries:

  1. Section 1 should be written without any subsection. In current form, it looks like a report. There is no need to put ‘structure of the paper’. It is well understood that last para of section 1 talks about the content of paper briefly. Authors may or may not agree.
  2. L181, authors may use the coordinate system in terms of N and E for clarity to the reader.
  3. There should be a space between numeral and corresponding unit. E.g. L228, it should be ‘1 kHz’ not 1kHz. Authors should check the complete manuscript.
  4. 3, authors should show the schematic of electrodes placement along with the working principle in brief.
  5. Some figures need to be replotted for clarity. E.g. Fig. 9 is not clear. How many iterations were carried out?
  6. 10c, can authors comment why x-axis has weird numbers, not a particular scale?
  7. Authors compared the reconstruction with LSTM is more than 1000 times faster. Can authors comment on difference in computational resources needed for LSTM compared to other methods? Does LSTM consist of any disadvantage in authors’ opinion?

Reviewer 3 Report

This manuscript introduces an EIT method for measuring brick wall humidity based on LSTM. The results are compared with artificial neural network (ANN) for both real and synthetic data. The whole paper is of interest and well prepared. I only have some minor comments for the authors to address. 

  1. It was mentioned in line 124 of page 3 that the Gauss-Newton method is a direct method but not an iterative method. This is confusing since Gauss-Newton generally needs iterations with given initials.
  2. The  in Table I are exactly the same as . Is it a typo here?
  3. The legends in Figure 6 are confusing since it is hard to see these markers.
  4. There is lack of details on the generation of 30000 training samples. Please add details and describe explicitly the difference of training samples with the testing samples.

Round 2

Reviewer 1 Report

I like Poland, I like Polish gothic architecture. I welcome the desire of the authors to preserve historical monuments. I welcome the author's approach to non-destructive monitoring of the state of old brick walls.

In my opinion, in the previous version of the manuscript, the authors overemphasized the algorithmic part of their research and omitted the physical foundations of impedance tomography and measurement techniques.

I think readers need to be confident that the authors have used reliable measurement methods, without having to read many of the authors' previous publications.

For this reason, I asked the authors a lot of questions from the reader's point of view.

To my surprise, the authors answer each question very carefully to me personally, but at the same time give the readers scant comments.

I am very sorry that readers cannot read the detailed explanations I now have.

I think the authors have made significant improvements to the current version of the manuscript, and it may be published.

However, I will ask the Editor to publish the authors’ reply to my comments as the supporting material to the main text of the paper.

Reviewer 2 Report

Authors have incorporated the suggestions. The manuscript may be accepted.